# Diversity matters in wheat mixtures: A genomic survey of the impact of genetic diversity on the performance of 12 way durum wheat mixtures grown in two contrasted and controlled environments

**Pauline Alsabbagh[1], Laurène Gay[1], Michel Colombo[1], Germain Montazeaud[2], Morgane Ardisson[1], Aline Rocher[1], Vincent Allard[3], Jacques L. David[1] ***

**1** UMR AGAP, CIRAD, INRAE, Institut Agro Montpellier, Université Montpellier, Montpellier, France, **2** Department of Ecology and Evolution, University of Lausanne, Lausanne, Switzerland, **3** UMR GDEC, INRAE, Université Clermont Auvergne, Clermont-Ferrand, France

\* jacques.david@supagro.fr

## Abstract

In ecology, an increase in genetic diversity within a community in natural ecosystems increases its productivity, while in evolutionary biology, kinship selection predicts that relatedness on social traits improves fitness. Varietal mixtures, where different genotypes are grown together, show contrasting results, especially for grain yield where both positive and negative effects of mixtures have been reported. To understand the effect of diversity on field performance, we grew 96 independent mixtures each composed with 12 durum wheat (*Triticum turgidum* ssp. *durum* Thell.) inbred lines, under two contrasting environmental conditions for water availability. Using dense genotyping, we imputed allelic frequencies and a genetic diversity index on more than 96000 loci for each mixture. We then analyzed the effect of genetic diversity on agronomic performance using a genome-wide approach. We explored the stress gradient hypothesis, which proposes that the greater the unfavourable conditions, the more beneficial the effect of diversity on mixture performance. We found that diversity on average had a negative effect on yield and its components while it was beneficial on grain weight. There was little support for the stress gradient theory. We discuss how to use genomic data to improve the assembly of varietal mixtures.

## Introduction

Maintaining crop yields will become increasingly challenging in the future due to the double constraint of an increased climate variability induced by climate change and a socio-economic context where both mineral inputs and phytochemicals use will tend to decrease [1–5]. For example, French wheat productivity measured in farms has reached a plateau since the 1990's [1, 6, 7], while the rate of genetic progress measured in non-limiting environments in

**Data Availability Statement:** All files are available from the database on the following link: https:// data.inrae.fr/dataset.xhtml?persistentId=doi:10. 15454/FG261F.

**Funding:** JD was funded by the EU Horizon2020 SOLACE (grant number 727247). The funder had no role in study design, data collection and analysis, decision to publish, or preparation of the manuscript.

**Competing interests:** The authors declare no competing interest.

experimental stations remained constant [4]. This strongly suggests that the genetic progress barely compensates for the progressive degradation of the pedo-climatic potential faced by farmers in natural fields and that, farmers practices are diverging from practices in experimental stations.

While successful genetic progress led to dramatic yield increases during the XX[th] century for many major crop species in intensive agricultural systems, the breeding strategy also contributed to an important decrease of the genetic diversity within and between fields [8, 9]. During the XX[th] century, it was estimated that 75% of crop diversity vanished from the farmers' fields at a global scale [10]. Reintroducing higher levels of diversity in cultivated fields has been proposed as a putative solution to stabilize production and restore other ecological services [11–13], in particular within the context of sustainable agriculture [14–17]. Besides strategies to associate several crop species that received much attention, intra-specific genetic diversity has also been shown to play an important role in ecosystem functioning and stability [18–20]. In particular, varietal mixtures have been proposed as a tool to increase crop genetic diversity without the need of extensive breeding efforts [21] by assembling several varieties of the same species within the same field [3, 5, 22–24]. In France, varietal mixtures are under a renewed attention by farmers and the practice rose from below 2% of the wheat surface in 2010 to 8% in 2018 [25] mainly due to their capacity to stabilize yield over years [25, 26].

Classically, agronomic and ecological research investigate diversity effects by measuring overyielding, which is defined as the difference between the yield of the mixtures and the average yield of their components grown in monoculture [27, 28]. Meta-analysis showed that, on average, over-yielding ranges between 3.5% for wheat [29], 3.9% for wheat and barley [12] and 5.4% for crop varietal mixtures [17]. Nevertheless this over-yielding is highly variable both between mixtures differing in composition and between environments. Negative over-yielding values have been observed in some situations [12, 17, 30]. The variability observed in mixture performance can find conceptual grounds in ecological theories. First, the hypothesis of niche complementarity proposes that niche differentiation between organisms (component varieties in the present example) optimizes the use of natural resources, leading to an enhanced productivity [31]. Facilitation between different cultivars [32] or barrier and dilution effects of pathogens [29], could also promote natural regulatory processes within the mixture [25] by reducing negative density-dependent effects of enemies [33] and improve the performance of the mixture. Oppositely, negative effects of diversity can be explained by the kin selection theory. This hypothesis states that communities with higher genetic relatedness on some "social" traits should be more fit than genetically unrelated and diverse communities for these traits [34, 35]. Mixture performance is also highly variable between environments, notably because of the environmental stress intensity. The ecological mechanisms acting in diverse communities, such as varietal mixtures could be less effective under optimal condition, in absence of biotic or abiotic stress than under stress [36–40]. Under this assumption, species redundancy in diverse communities are able to maintain functioning during increased environmental stress. Tolerant species within the mixture provides a greater guarantee to maintain functioning by compensating the failure of sensitive species [41]. Facilitation and competition between plants varies along environmental gradients with clear positive interactions dominating under stressful conditions [36]. In the specific case of varietal mixtures, such an effect has been observed through a clear over-yielding increase along a biotic stress gradient [29]. However, in that study, abiotic stress variations did not produce any significant effect on over-yielding.

Given this observed high variability, understanding the mechanisms underlining mixture performances is necessary to help farmers design efficient mixtures. If a few assembly rules have been proposed in the specific context of disease control [26], the identification of key criteria for mixture assembly is still necessary [42–44], in particular to design mixtures adapted

to abiotic stresses. The recent progress of genomics and high-throughput genotyping that give access to allelic variations along the genome at a moderate cost [45, 46] opens a great opportunity to understand the effect of genetic diversity on mixture performance at the gene level. This could then be used to design criteria for mixture assembly. For example, a positive relationship between the diversity over a single 310 kb genomic region was found associated with an increase of productivity of *Arabidopsis thaliana* two-component mixtures [47]. Identically, another study on mixtures of genotypes of Arabidopsis thaliana, showed a SNP on the chromosome 2A which has a higher size effect in polycultures than in monocultures [48]. Oppositely, genetic diversity at a single locus on chr 6B was negatively associated with grain yield in two-component mixtures of durum wheat [49]. These two examples illustrate the complexity of the mechanisms through which within-field plant genetic diversity in particular, may be beneficial or detrimental to crop performance. Majority of the studies tackling varietal mixtures, as these two studies are based on two-way balanced mixtures. Yet, the mixtures with highest interest for agronomical perspectives will likely be more complex, since the number of species has been identified as a key criterion affecting ecosystem performances [50].

To go beyond the very large combinatory of multiple ways mixtures, we propose here to work on complex mixtures with high genetic diversity, with an explicit focus on the effects of diversity at a large number of loci on mixture performance. We report the results of an experiment where we built 96 mixtures, each made up of 12 independent durum wheat lines (*Triticum turgidium* subsp. *durum*) chosen within a panel of 96 lines. Given the complexity of the experimental design, individual lines were not observed in monoculture and over-yielding could not be computed. The objective of the study was not to demonstrate the agronomic interest of mixtures that is now acknowledged, but rather to explore a large range of allelic variation on several thousands of markers and seek for association between the phenotypic values of major agronomical traits and within-mixture genetic diversity. To this aim, we extended the classical GWAS approach to a Genome Wide Diversity Association studies (GWDAS). Furthermore, in order to test the stress-gradient hypothesis, these associations were studied in two contrasted conditions for water availability. In durum wheat, drought is one of the main causes of yield losses that can vary from 10 to 80% depending on the year, in the Mediterranean region [51]. We therefore explored how the effects of genetic diversity were affected by drought, a major abiotic stress, using the Pheno3C INRAE phenotyping platform at INRAE Clermont-Ferrand.

## Materials and methods

### Mixtures design

Ninety-six inbred lines of durum wheat (*Triticum triticum durum* Th.), were chosen from a set of 180 inbred lines, derived from an evolutionary pre-breeding population, hereafter EPO, developed at INRAE Montpellier, France [49–53]. These lines have been described for a large number of traits, and exhibit high phenotypic variability for both above- and belowground traits [52], and harbour interesting major resistances to foliar diseases [54]. We discarded extreme tall and short components to reduce the competition among lines within plots, due to plant height [52, 55, 56]. Among the 180 lines, we selected 96 lines to maximize homogeneity in stem height (90–110 cm, S1 Fig) and restrict flowering date variation between lines to 5 days (S2 Fig).

Ninety-six 12-way mixtures were prepared by assembling for each of them a number of 70 seeds from twelve distinct durum wheat lines, drawn from the 96 selected genotypes. Together, the 96 mixtures represent an incomplete balanced design. No mixture had the same composition and each line was present in 12 mixtures with an independent set of neighbours. The

design was realized by a homemade R script. The rationale was to sample mixtures iteratively, sampling at each step new mixtures in the remaining available lines and verifying the unicity of each mixture.

## Field experiment

The 96 mixtures were sown in Clermont-Ferrand, France, at INRAE Crouël (45˚78' N, 3˚08' E, 401 m a.s.l), in October 2017 on the Phéno3C field-phenotyping platform (INRAE PHACC Experimental Unit). Phéno3C is installed on a browned fluvisol with predominantly clay-loam soil. The surface horizon has a pH of 8.04, a percentage of organic matter content of 3.5, and a depth of 90 cm, its water reserve is 260 mm. This platform offers unique conditions to contrast in the same site, controlled water deficit (CWD) and rainy (R) conditions, via a system of rain-out mobile shelters, to generate the required level of water stress. Mixtures were sown in 8-rows plots of 2.185 m$^2$ each (1.15 m width, 1.9 m length and 17 cm between rows), in four blocks in each R and CWD conditions (48 mixtures in each). None of the 96 mixtures is identical to the other. The idea is not to choose the best mixture or how a mixture behaves differently under different stresses. The purpose of the mixtures is to see the effect of diversity on yield and whether it changes under stress or not. The trial was managed following local agronomic practices. The sowing was realized on 15/11/2017, and the harvest on 03/07/2018 for CWD and on 10/07/2018 for R plots. The 48 mixtures under CWD were covered by automatic rain-out from 02/02/2018 to create a controlled water deficit. The 48 mixtures under R, were maintained under the local climate and no irrigation was needed.

## Measurement of soil water content

To measure the soil volumetric water content (SWC), sensors of CS 655 (Cambell Scientific, Logan, UT, USA) were installed at four depths (10, 35, 50 and 75 cm) at three locations per plot to record the volume hourly. Total soil water content was calculated by integrating sensor values up to 100 cm depth (average soil depth of the plots). Average soil field capacity and available water capacity are respectively 570 and 280 mm. All other environmental conditions were identical between the two regimes.

## Phenotypic data

**Traits recorded at the plot level.** Heading date (HD) was measured when 50% of the spikes headed. The number of spikes per meter square (NSM2) was determined after anthesis, by manual counting of spikes on a 1 m long transect in the central row. After harvest, the yield of each mixture (Y) was computed as the raw weight of harvested kernels, measured at a 0% of humidity (humidity-meter, TM, Tripette and Renaud, France). It was corrected according to the number of some missing rows due to cropping aleas. Y is finally given in grams per m$^2$. The whole plot harvest was divided by half with a seedburo divider followed by second division by 8 using a precision Retsch divider. A final count of 1200 kernels from a sample was done on a Contador device (Pfeuffe) and then precisely weighed. This is how we obtained a representative sampling of 1200 kernels from each plot. The Number of Kernels per meter square (NKM2) was calculated from Y and TKW as NKM2 = 1000 x Y/ TKW. The number of kernels per spike (NKS) was calculated as NKS = 1000xY/(TKWxNSM2).

**Traits recorded on 10-individual spikes per plot.** Ten stems with fully developed spikes (stems with aborted spikes were discarded) were collected randomly from the central row in each mixture/plot before harvest. Stems were cut at the soil level. We measured the stem length (SL, cm) and the spike length (SpL, cm). These values were then averaged over the 10 stems to obtain one value per mixture.

## Genotypic data

Genotypic data for the 96 EPO lines was obtained in a previous study using the high-through-put genotyping array TaBW280K [46], as detailed in Ballini (2020) [54].

The physical positions of the SNPs were estimated by blasting first on the reference genome SVEVO [57] and confirmed by linkage disequilibrium analysis realised using home-made scripts. We only kept the SNPs for which we had a hit on the genome sequence by Basic Local Alignment Search Tool (BLAST). The genotyping data resulted in 96,562 polymorphic SNPs after filtering with a minimum of 95% of similarity. Allele frequencies of SNPs in each of the 96 12-way mixtures were estimated assuming a balanced contribution from each of the 12 lines:

$$Fi, j = \frac{1}{12} \sum_{k=1}^{12} G_i^{k,j}$$

where Fi,j is the allele frequency of the i[th] SNP (from 1 to 96,562) in the j[th] mixture (j from 1 to 96) and $G_i^{k,j}$ is the genotype of the k[th] (from 1 to 12) line used in mixture j at locus i, and can be 2, 1 or 0.

All seeds were produced in the same trial in single genotype plots, which reduces the between environment maternal effect and provide an homogeneity in seed quality. The allele frequencies in each mixture at sowing is taken as the best proxy of the relative proportions of the different genotype at the adult plant stages.

We measured genetic diversity of each mixture at each locus using the Nei's heterozygosity [58] as follows:

$$HE_{i,j} = 2F_{i,j}\left(1 - F_{i,j}\right)$$

Because diversity is a function of allele frequency, it is highest when allele frequencies are balanced. There is therefore a risk to detect an effect of diversity that is in fact an effect of allele frequency, for example, when a favourable allele is rare. We kept the 31,059 SNPs whose minimum frequency in the mixtures was below 0.5 and for which the maximum frequency was above 0.5. In addition, we imposed that the difference between these minimum and maximum values be at least 0.5.

## Statistical analysis

All analyses were performed on the software R.

## Preliminary analyses

To evaluate the impact of the two water regimes on each trait, a simple ANOVA declaring treatments and blocks was analysed. To determine the relationship among traits and the influence of the water regime on these relations, we performed a matrix of Pearson correlation [59] on the whole data set (All treatments) declaring a treatment effect, and on the two water regime data sets separately.

## Genome Wide Genetic Diversity Association (GWDA) and Genome Wide Frequency Association (GWFA)

The genome wide genetic diversity association GWDA, respectively the genome wide frequency association GWFA, analysis were built by doing regression analyses on the diversity index HE, resp, the allelic frequency F. We also declared the water regime effect and its

interaction with both genetic variables (F and HE). We included a block effect to account for the experimental design. This resulted in the following model(s) for each SNP:

$$p = \mu + Block_T + T + HE_k(F_k) + TxHE_k(TxF_k) + e,$$

where P is the phenotypic trait in each mixture, T the effect of the water regime (CWD or R), $Block_{/T}$ the block nested in the treatment T effect. $HE_k$ (GWFA) and $F_k$ (GWDA) are respectively the regression slopes of the genetic diversity, resp. the allele frequency at the $k^{th}$ SNP locus, and $TxHE_k$ (or resp. $TxF_k$) the interaction effect between the treatment and genetic diversity (or resp. allelic frequency). Treatments, diversity and interaction effects were assessed using a sequential type 1 test with the least square method. Effect were tested in the following order (Block, T, He $_k$, resp. $F_k$) and $TxHE_k$ (resp. $TxF_k$). P-values were–log10 transformed for clarity as in a GWAS study. For each locus for which we found a significant association with HE, we compared the GWDA to the GWFA one by an anova test (thereafter called the "HE vs. F test"). Its significance threshold was determined by a Bonferonni correction to account for independent detection of multiple QTL.

For the QTLs found associated to the interaction between HE and the environment, we calculated the correlation between the allele frequency of the associated QTL and the trait in each environmental condition separately.

**Significance threshold.**   The threshold for significant association, i.e., for declaring a HE-QTLs (with GWDA) and F-QTLs (with GWFA) was determined using the Galwey method [60]. Briefly, a matrix of linkage disequilibrium among polymorphic SNPs was used to estimate the effective number of independent tests. We found 1774 independent segregating chromosomes fragments through the method. A corrected Bonferroni threshold for was then 0.05/1774, giving a–log10(*PValue*) threshold of 4.55.

**Quantitative trait Loci (QTL) boundaries.**   A linkage disequilibrium based method inspired from [61] was used to define QTL boundaries for each SNP showing a significant signal of association. For each chromosome and each trait, we computed the linkage disequilibrium (LD) among significant markers, using Hill and Robertson's $R^2$ estimator [62]. These LDs were square roots transformed to approximate a normally distributed random variable [63]. Then markers were clustered by LD blocks. Clustering was realized by average distance using a cutoff of 1-"critical $R^2$". Critical $R^2$ ($R^2c$) was defined as the 99.9$^{th}$ percentile of the distribution of unlinked $R^2$ computed between pairs of markers randomly sampled from different chromosomes. This threshold accounts for a risk of 0.1% to be in LD by chance. QTL boundaries were finally defined as the minimum and maximum map position of significant markers belonging to the same LD block. The most significant marker was used to represent QTL size effect and minor allele frequency (MAF). QTL of different traits were considered to overlap when they had at least one common significant marker and were located at a physical distance below one tenth of the total physical length of the chromosome, as presented in [61].

## Co-localization of QTL with published QTL

Using the Triticum turgidum Durum Wheat Svevo (RefSeq Rel. 1.0)—GrainGenes (usda.gov) and the table of the meta-QTL analysis [64], our QTLs were co-located using their physical location in bp on the SVEVO genome of durum wheat using the interval defined beyond [57].

## Global genomic approach of water regime impact

To study the impact of genetic diversity at the genome level, we aggregated the effects of diversity across loci for each water regime and for each trait. Only SNP markers fitting the

conditions for HE computations (see Genotypic data) for both treatments were kept, only 31,059 were left. We first fitted the following model for each SNP and every trait in each treatment (CWD and R):

$$P = \mu + \beta_k HE_k + Block + e,$$

where P is the response trait in each plot (from 1 to 48, either in R or in CWD), $\beta_k$ is the slope of the partial regression of the genetic diversity (HE) at locus k (k from 1 to 13,059) on the trait, and Block are the blocks in the considered treatment.

We obtained for each SNP a pair of regression slopes for the effect of HE on the trait, both in R and in CWD. We counted the number of positive or negative HE slopes for each trait and each water treatment. We assessed whether diversity had a rather positive or negative effect on the trait by testing whether there was a significant difference in the number of positive and negative slopes by a Wilcoxon test. In a second step, we examined the stress gradient hypothesis (HE has a more positive effect in stress environment, i.e., higher under CWD than under R) by testing whether the slopes under CWD were on average higher than under R, using a paired Wilcoxon test on the paired differences in slopes computed across treatments for each SNP.

All data and scripts can be found at https://doi.org/10.15454/FG261F/I7OJKU.

## Results

### Water stress

The soil water reserve in the R plots was never significantly below the limit of "easily accessible water" during the whole crop cycle leading to infer that plants in R condition did not experience water stress (Fig 1). In the CWD, from the 7th of April (booting stage) to the end of the plant cycle, the water reserve fell below the threshold (400 mm) of the easily accessible water and reached extremely low levels during the last part of the cycle.

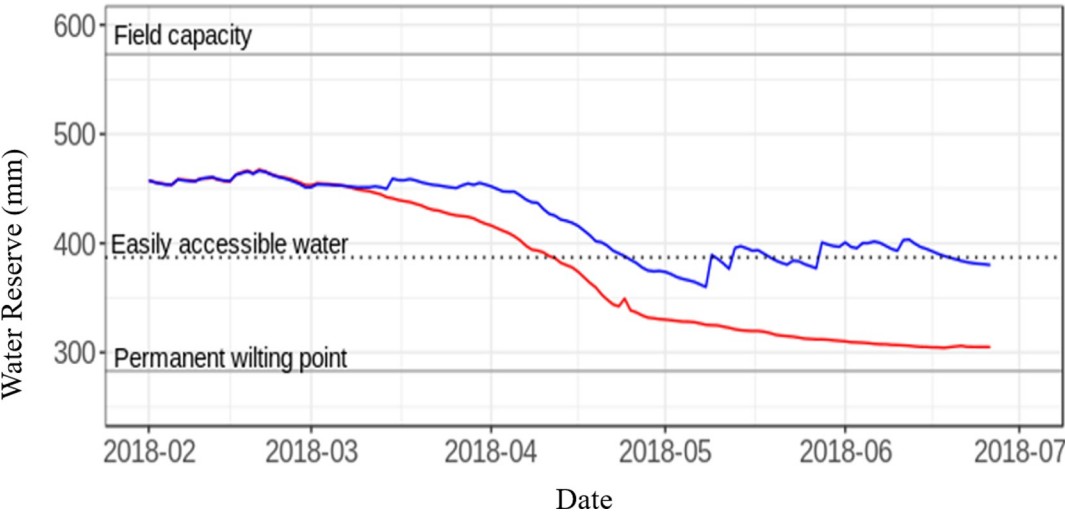

**Fig 1. Dynamic of the cumulative rainfall from sowing and of the soil water reserve Pheno3C for the 2018 campaign, in the two treatments.** Horizontal lines figure the soil water reserve (in mm) at Field Capacity Easily accessible water and permanent wilting point. Blue line is for plots under rainy conditions (not sheltered). Red line is for plots under controlled water deficit conditions (under shelter from the end of February).

## Impact of controlled drought on phenotypic traits

CWD had a significant effect on all traits studied, except SpL (Table 1). Under stress, on average, plants flowered 5 days earlier. Plant morphology was affected by stress leading to a reduction of plant height by nearly 10%. In term of yield elaboration, water stress reduced significantly all yield components. The impact on TKW (-7.63%), NSM2 (-8%) and NKS (-10.3%) was moderate but severe on NKM2 (-25%). These effects are in the expected direction since it is known that stress reduces performance.

## Correlation between agronomic traits

Under both conditions, Y was correlated to NKM2 and NSM2, and NKM2 was correlated to NSM2. However, water condition played a big role in traits relationships. For instance, plots having longer stem were more productive in the R condition (r(SL-NKM2) = +0.3, p-value = 0.03) but under CWD we did not find significant correlation between stem length and productivity (-0.12, p-val = 0.4). Relationship between NKM2 and TKW was also affected by the water regime. Under water stress, the correlation was not significant but significantly negative (r = -0.43, p-value = 0.002) under rainy conditions (Fig 2). This indicates that water stress induced a compromise between grain size and grain number.

## Genome wide association studies

**Associations with allelic frequencies (GWFA).** Six F-QTLs involving SNP frequencies were detected from 604 SNPs with–log10(PValue) higher than the threshold. Traits involved were SpL and TKW (S1 Table). SpL has the highest number of significant associations in GWFA (5 F-QTLs), while only one F-QTL was found for TKW (S1 Table). The F-QTL of TKW, located on the chromosome 2A had the most significant association with a–log10 (PValue) of 5.77 (S3 Fig and S1 Table).

**Genotype (F) x Environment.** Pvalues of Treatment x Frequencies were rarely significant except for one F-QTL (AX-89757725,–log10(PValue) = 5.14) associated with TKW, located at 546,024,925 bp, within the interval [545,597,462; 547,234,231] on the chr 7B. For this QTL, the effect of F was low in R but had a strong effect in CWD. The frequency variation range for this F-QTL was not large. The increase in frequency was favourable under water stress conditions and neutral in rain-fed conditions.

**Table 1. Average values in durum wheat mixtures between two water regimes for 8 traits.**

|  | Traits | R | CWD | Average Mean difference (%) | -log10 (PValue) |
|---|---|---|---|---|---|
| Average of 10 individual spikes | SL | 74 | 67 | -9.45 | > 15.65* |
|  | SpL | 6.9 | 6.9 | 0 | 0.39 |
| Plots value | NSM2 | 294 | 272 | -8 | 1.6* |
|  | NKS | 34.69 | 31.09 | -10.3 | 2* |
|  | NKM2 | 10560.95 | 8441.714 | -25 | 5.6* |
|  | TKW | 53.43 | 49.35 | -7.63 | 15.21* |
|  | Y | 266,5 | 204 | -24 | 10.10* |
|  | HD | 127 | 122 | -4.75 | >15.65* |

CWD: Controlled water deficit conditions, R: Rainy conditions, SL: Stem length in cm, SpL: Spike length in cm, NSM2: Number of spikes per m$^2$, NKS: Number of kernels per spike, NKM2: Number of kernels per m$^2$, TKW: Thousand kernel weight in g, Y: Yield in gr/m$^2$, HD: Heading date: days after 1/1/2018, Average Mean difference (%): the percentage of difference in the trait between the two conditions. The threshold of–log10(PValue) is 1.301.

* Significant test.

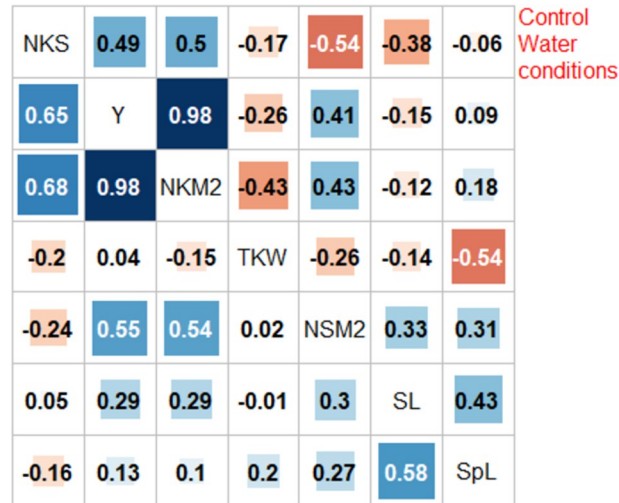

**Fig 2. Pairwise Pearson correlation coefficients (r) between functional traits for the 48 mixtures under controlled water deficit conditions (above the median) and the 48 mixtures under rain-fed conditions (below the median).** Positive correlations are in blue and negative in red. The darker and larger the square, the higher the correlation between traits. Traits have been clustered according to the *hclus* procedure of the *corrplot* R package function. SL: Stem length in cm, SpL: Spike length in cm, NSM2: Number of spikes per m², NKS: Number of kernels per spike, NKM2: Number of kernels per m², TKW: Thousand kernel weight in g, Y: Yield in gr/m².

**Association with diversity (GWDA).** From the 31K SNPs considered, 211 showed a significant signal of association involving 6 traits. These 211 SNPs were grouped in 30 HE-QTLs on Y, NKS, NKM2, TKW, SL and SpL (S2 Table). It is striking to see that the number of traits involved in the associations with SNP diversity is higher than in GWFA. To ascertain that these associations with HE were not due to the overfitting of a simple linear regression based on F, we compared by the *anova* function of R, the HE and F models for the 30 HE-QTLs, (HE vs. F test) with a significance threshold of 0.001 (i.e., 0.05/30) or a–log10(PValue) = 2.78. Four HE-QTLs were left significantly more associated with HE than with F (Table 2): one HE-QTL associated with the NKM2 (–log10(PValue(HE vs F)) = 3.96) on the chr 1B, with a negative effect of HE; one HE-QTL for the NKS (–log10(PValue(HE vs F)) = 4.99) on the chr 1B with a negative effect (Fig 3), who is the same HE-QTL found for the NKM2; one HE-QTL for the TKW (–log10(PValue(HE vs F)) = 3.19) on the chr 4A, with a positive effect (Fig 4) and one HE-QTL for SpL (–log10(PValue(HE vs F)) = 3.21) on the chr 3B, with a positive effect. For the 26 others

**Table 2. HE-QTLS Significant associations between genetic diversity and traits.**

| Trait | SNP | Lower bound | Peak | Upper bound | Chr | GWDA–log10(PValue) | HE vs F test–log10(PValue) | r |
|-------|-----|-------------|------|-------------|-----|--------------------|-----------------------------|---|
| SpL | AX-89676059 | 4186235 | 4459123 | 4493159 | 3B | 4.61* | 3.21* | + 0.37* |
| NKS | AX-89672881 | 32910445 | 32945067 | 32946674 | 1B | 4.99* | 4.99* | -0.41* |
| NKM2 | AX-89411835 | 32910445 | 32910375 | 32946674 | 1B | 4.94* | 3.96* | - 0.28* |
| TKW | AX-89377854 | 635466248 | 635479909 | 635479979 | 4A | 4.92* | 3.1* | + 0.29* |

* Significant–log10 (PValue). SpL: Spike Length in cm, NKS: Number of kernels per Spike, NKM2: Number of kernels per m², TKW: Thousand Kernel Weight, SNP: Single Nucleotide Polymorphism, Peak: physical position of the SNP on the chromosome in base pairs (bp), Lower bound: the minimal physical position that can have the SNP on the chromosome, Upper bound: the maximal physical position that can have the SNP on the chromosome, Chr: chromosome, QTL: Quantitative Trait Loci, r: value of the correlation between the genetic diversity of the QTL and the trait. The threshold of GWDA–log10(PValue) is 4.46. When GWDA–log10(PValue) is > 4.46 the association between the trait and the genetic diversity of the QTL is significant, and vice versa. The threshold of HE vs F test–log10(PValue) is 2.78. * Significant test.

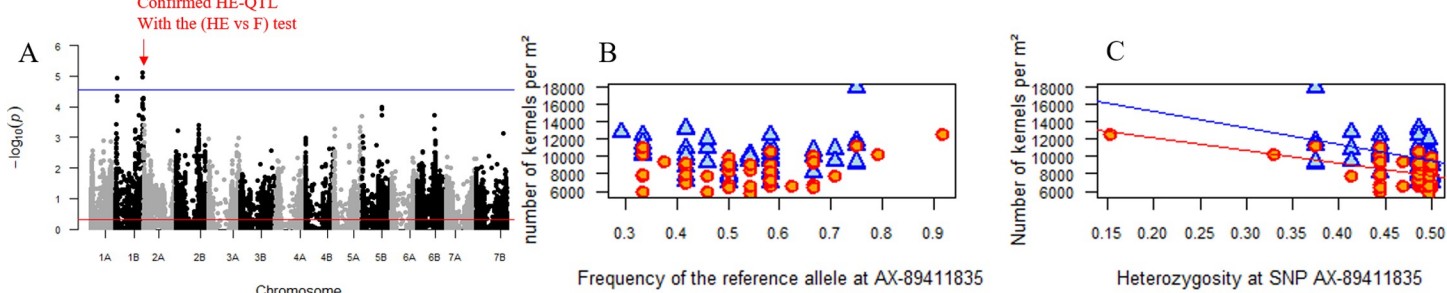

**Fig 3. Association between the number of kernels per m² (NKM2) and genomic diversity.** (A) Manhattan plot reporting p-values (-log10 transformed) for the association tests between NKM2 and diversity index (Nei's diversity) at 31k eligible SNPs distributed along the durum wheat genome (GWDA). The solid blue line represents the Family-Wise Error Rate (FWER) of 5% computed with the Galwey method. The SNP "AX-89411835" located on the chromosome 1B has passed the He vs. F test (see text). (B) Association between of the 96 allelic frequencies of the SNP "AX-89411835" in the mixtures and their NKM2 values. Red points correspond to control water deficit plots, blue points to rain-fed plots. (C) Relationships between the genetic diversity of the SNP "AX-89411835" and NKM2. Lines are the slopes of regression. The slope for the controlled water deficit treatment has been computed without the outlier point on the x-axis.

HE-QTLs, HE did not significantly improve the model compared to F, even though no significant associations were detected with F. This may be due to a lack of power and the distribution of allele frequencies. This approach is certainly over-conservative, but it ensures that associations detected with HE had a reduced probability to be false positives due to overfitting.

*Genotype (HE) x Environment.* For Y and NKM2, we found one common significant HE-QTL, located at 7,275,378 bp on the chr 2A with varying effect depending on the water conditions. HE had a non-significant negative effect under R, but a significant positive effect under CWD (Fig 5). This HE-QTL of the G x E interaction remained significant after the HE vs. F test for Y (–log10 (PValue) = 3) but not for NKM2 (-log10 (PValue) = 0.1).

**Co-localisation.** For the GWFA, co-localization with already published QTL were found for almost all the QTLs detected (S3 Table), with a varying number of matches. The 5 SpL F-QTLs co-localised with QTLs associated to the spike characteristics (Unpublished from [64]); [65, 66], to plant height [67, 68], and yield components traits related to the number of kernels [64] and the weight of kernels [69, 70]. The TKW F-QTL co-localised with 5 published QTLs associated to grain quality and weight [71–73] and to heading date [67].

The HE-QTL found for TKW on chr 4A co-localized with QTLs associated to the spike length (unpublished data from [64], plant height [74], grain yield [74], root characteristics [75]

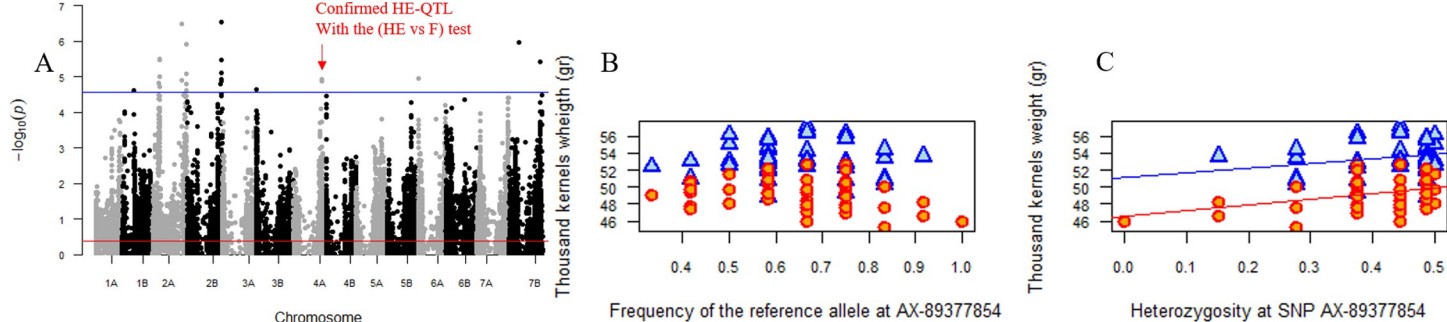

**Fig 4. Positive impact on thousand-kernel weight (TKW) of the genetic diversity at the SNP "AX-89377854" located on the chromosome 4A.** (A) Manhattan plot reporting p-values (-log10 transformed) for the association tests between TKW and diversity at 31k eligible SNPs distributed along the durum wheat genome (GWDA). The solid blue line represents the Family-Wise Error Rate (FWER) of 5% computed with the Galwey method. (B) Association between the allelic frequency of the SNP "AX-89377854" and TKW. Red points corresponds to plots under controlled water deficit conditions. Blue points corresponds to plots under rain-fed conditions (C) Relation between the genetic diversity of the SNP "AX-89377854" and TKW. Lines represent the slopes of the regression.

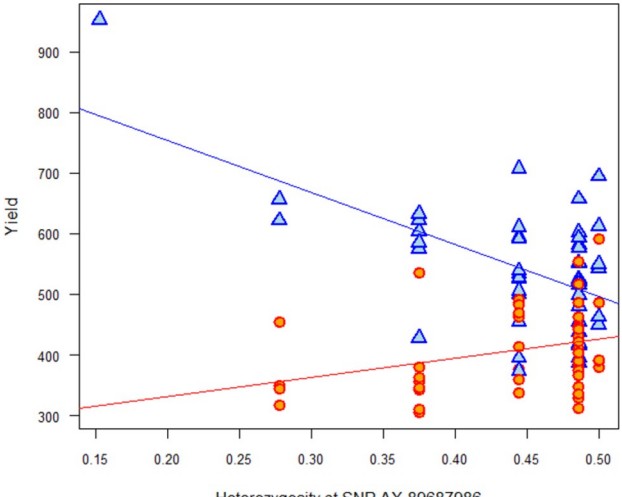

**Fig 5. Contrasted impact of the genetic diversity at locus "AX-89687986" on the yield (Y) on the 96 durum wheat mixtures under two water regimes: In red, positive effect of diversity under controlled water deficit conditions (red points and line) and in blue negative effect (ns) under Rain-fed conditions.**

and biomass [74] (S4 Table). The HE-QTL found for SpL on the chr 3B co-localized with QTLs associated to the grain yield [74], heading date [76] and root characteristics [75]. The same HE-QTL found for NKM2 and NKS on chr 1B col-localized with grain yield [74], thousand kernels weight [69] and test weight [71].

For the interaction of the significant (F-QTL of the TKW) x Water regime, we found 4 colocalized QTLs in the literature on the number of kernels/spike, the grain yield per plant, the thousand kernels weight [77] and test weight [67].

**Global genomic impact of diversity.** The HE slopes on traits were distributed between positive and negative values (e.g, for NKS, Fig 6). The Wilcoxon tests suggested a significantly larger number of SNPs for which diversity was positively associated with stem length, in both

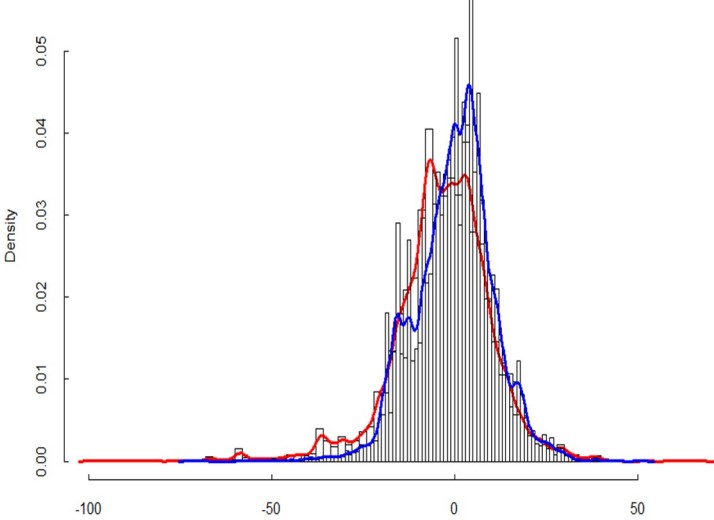

**Fig 6. Slopes distribution of NKS under each environmental condition separately.** Blue line for plots under rain-fed conditions. Red line for plots under controlled water deficit conditions.

**Table 3. Number of SNPs with a negative and positive effect on traits under CWD and R conditions.**

| Trait | # of negative slopes in CWD | # of positive slopes in CWD | –log10 (PValue) Wilcoxon Test in CWD | # of negative slopes in in R | # of positive slopes in R | –log10(PValue) Wilcoxon Test in R |
|---|---|---|---|---|---|---|
| SL | 5501 | 7587 | >15.65* | 5547 | 7541 | >15.65* |
| SpL | 6689 | 6399 | ns | 5289 | 7799 | >15.65* |
| NSM2 | 6445 | 6643 | ns | 6343 | 6745 | ns |
| NKS | 7434 | 5659 | >15.65* | 6666 | 6427 | 7.05* |
| NKM2 | 7349 | 5739 | >15.65* | 6742 | 6346 | ns |
| TKW | 5703 | 7385 | >15.65* | 5958 | 7130 | >15.65* |
| Yield | 7070 | 6018 | >15.65* | 6793 | 6295 | ns |

#: Number, CWD: controlled water deficit conditions, R: rainy conditions, SL: Stem length in cm, SpL: Spike length in cm, NSM2: Number of spikes per m2, NKS: Number of kernels per spike, NKM2: Number of kernels per m2, TKW: Thousand kernel weight in g, Y: Yield in gr/m$^2$. Wilcoxon test gives the -log10(PValue) of the SNP pair ranked Wilcoxon test. The significance threshold of the Wilcoxon test is 1.301.

* Significant test, ns: non significant test.

the stress (CWD) and normal (R) treatments (Table 3). This suggests that plants in diverse mixtures tended to grow taller. For yield and its components such as the number of kernels produced (Y, NKM2, NKS), we found a clear excess of SNP whose diversity had a deleterious effect in the stressful environment, but an excess of SNPs with positive effect or not significant effect in the R treatment. It was the reverse for the grain weight (TKW), where diversity had a positive effect for most SNPs (Table 3). Interestingly, while the spike density per m$^2$ (NSM2) is one of the most important yield component, no clear global genetic impact was detected.

## Test of the stress gradient hypothesis

For the traits related to fitness, the stress gradient hypothesis predicts a global positive change in the slopes values under stress compared to the non stressed conditions. The paired Wilcoxon tests was used to explore the changes in the HE slopes from R to CWD at the genome level for the 13K selected SNP (Table 4). The test was significant for most traits except NSM2 and TKW. We found that diversity increased plant height only slightly more under R than under CWD (Table 4). The situation was reversed for yield and its components (NKM2, NKS), where global genetic diversity was more detrimental under stressful conditions than under normal conditions.

## Discussion

In the present paper, we investigated the effect of genetic diversity on the performances of a population (here varietal mixtures) under contrasted stress conditions. First, the regression approach was extended to detect the effects of diversity *per se* at any given locus (GWDA). Second, the impact of diversity at a broad genomic scale was also investigated by cumulating results across loci. We detected effects of diversity at four loci (HE-QTL) on several phenotypic traits. Concerning the traits linked with yield and its main components, that can all be considered as proxies of plant fitness, significant HE-QTL were negatively linked to performance and more globally, an excess of loci with a negative impact of diversity *per se* at the genomic scale were identified.

Studies on crop mixtures are still scarce [78–80] and the present experiment is among the few [49] that explored the question using complex mixtures and by explicitly taking into account diversity at the genomic level. In studies or meta-analysis that compare a large number

**Table 4. Difference number and effect of the positive and the negative sum of the difference of slopes in CWD and in R (delta) and the p-value of their wilcoxon's test.**

| Trait | $\Delta > 0$ | $\Delta < 0$ | Wilcoxon test |
|---|---|---|---|
| SL | 6551 | 6537 | 6.58* |
| SpL | 5956 | 7132 | >15.65* |
| NSM2 | 7042 | 6046 | Ns |
| NKS | 6324 | 6769 | 13.54* |
| NKM2 | 5708 | 7380 | >15.65* |
| TKW | 6510 | 6578 | ns |
| Yield | 5756 | 7332 | >15.65* |

Abbreviation:

SL: Stem length in cm, SpL: Spike length in cm, NSM2: Number of spikes per m2, NKS: Number of kernels per spike, NKM2: Number of kernels per m2, TKW: Thousand kernel weight in g, Y: Yield in gr/m$^2$.

$\Delta > 0$, resp., $\Delta < 0$, is the number of SNP for which $\Delta_i = \beta_{CWD,i} - \beta_{R,i}$, is positive, resp. negative, $\beta_{CWD,i}$ and $\beta_{R,i}$ being the slopes of the regression of a trait on the diversity at a given SNPi under control water deficit (CWD) and rainy (R) conditions. Wilcoxon test gives the -log10(PValue) of the SNP pair ranked Wilcoxon test. The significance threshold is 1.301.

* Significant test. ns: non significant test.

of mixtures with their individual components, a small but positive overyielding is generally observed. Nevertheless these results also demonstrate the large variability observed between mixtures [29] that goes from clear yield enhancement [79], to significant negative effect [52]. In the present experiment, overyielding was not estimated since our focus was on the factors governing the variation of mixture performances, with the aim to better understand the variation observed for agronomic performance among mixtures and environments.

Our approach is close to other studies using allelic frequencies at many locus on natural populations. These approaches, as the genome-environment association (GEA) analysis often use the Pool-Seq approach in the aim to uncover SNPs associated to climatic or plant community variation [81–83]. The diversity effects at the genomic level have been reported previously [47, 52, 49], in experiments in which neighbouring genotypes were grown in spatially alternative rows. Here, we report on a completely random neighbourhood among genotypes in highly complex mixtures.

## The challenge of distinguishing the effect of diversity from the effect of allele frequency

Exploring associations between genetic diversity *per se* and phenotypic traits in mixtures, raises a number of methodological challenges and in particular the risk of model overfitting. The regression approach used in GWAS analysis is sensitive to false positives due to the large number of tests performed. Using HE as a regressor further increases this risk because the model can overfit the data if there is a direct association between allele frequency and a trait, since it permits to fit non-linear relationships between allele frequencies and the trait. If the GWDA is regressing on one predictor (HE), as in GWFA (F), it is not yet clear if statistical bias could generate an excess of false associations or if it better take into account plant x plant interactions in a heterogeneous population since it permits to fit non-linear relationships between allele frequencies and the trait. Investigation using simulations would be necessary.

By definition, HE depends on allele frequencies and if two homozygous genotypes have contrasted phenotypic values on average, it is difficult to disentangle the phenotypic effect due

to a change in the frequency of the favourable allele from a change of genetic diversity *per se*. For this reason, we were particularly cautious to ascertain our HE-QTLs and we reduced the number of significant associations from 30 to 4, by adding a filter on found HE associations, to be sure that the detected effects were significantly due to the diversity *per se* and not to allelic frequency (tests HE vs. F). It was surprising that the 26 remaining HE-QTL were not detected as F-QTL but this could be due to a non linear association of trait with F or a non-even distribution of allelic frequencies around 0.5, or to a lack of power. By design, the allelic frequencies in each of our 96 mixtures made of 12 genotypes had a reduced between-plot genetic variance compared to classical GWAS performed using individual inbred lines, hence reducing the statistical power [64]. It also reduced the variation compared to association studies based on pairwise comparisons [52].

Concerning the global genomic and non-parametric approach, we drastically reduced the number of SNPs for which frequencies varied largely among the 96 mixtures to explore correctly the effect of diversity *per se*. As we did not prune the SNP to keep only independent SNP, a level of redundancy shall exist. But as the Wilcoxon tests were highly significant for most traits, including yield, we confirm that our results are reasonably solid but deserve to be verified on repeated experiments involving more mixtures in larger sets of environments.

## Durum wheat mixture performances are impacted by their genetic composition

Despite the limits detailed above, our results suggest that genetic diversity had a detectable effect on mixtures performances. We are confident about our experimental design and the detected associations for spike length and TKW since they co-localized with QTLs published for the same categories of traits. This shows that the method we used to avoid false positives was efficient and also strengthens our confidence in the results of the four HE-QTLs.

**Diversity effect on mixture performance.** Table 5 summarizes the main observations on plant height, yield and its components and grain quality. HE-QTL were detected on important traits in each of these categories. The HE-QTLs on the number of kernels per spike (NKS) and on the number of kernels per $m^2$ (NKM2), two important components of yield, revealed a

**Table 5. Synthesis of the impact of diversity on yield and its components.** Stress: ↓ significant negative effect of the stress. F and HE-QTL: significant associations with the allelic frequency or the diversity. Global effect: + (resp. -) reports a significant excess of SNP whose diversity had a positive (resp. negative) effect on the trait. Stress gradient hypothesis: "Yes" if the number of SNPs that have positive impact of their diversity in the traits under CWD is significantly more than the number under R, "No" in the reverse case, ns: non-significant.

| | Stress | F-QTL | HE-QTL | Global effect | | Stress gradient hypothesis |
|---|---|---|---|---|---|---|
| | | | | **R** | **CWD** | |
| SL | ↓ | | | + | + | Yes |
| SpL | | 1B 2B 4B 5A | 3B (+) | + | | No |
| NSM2 | ↓ | | | | | ns |
| NKS | ↓ | | 1B (-) | - | - | No |
| NKM2 | ↓ | | 1B (-) | | - | No |
| | | | 2A GxE | | | |
| TKW | ↓ | 2A | 4A (+) | + | + | ns |
| | | 7B GxE | | | | |
| Y | ↓ | | 2A GxE | | - | No |

R: Rainy conditions, CWD: controlled water deficit conditions, SL: stem length, SpL: spike lenght, NSM2: number of spikes per $m^2$, NKS: number of kernels per spike, NKM2: number of kernels per $m^2$, TKW: Thousand-kernel weight in g, Y: yield in $g/m^2$.

negative effect of diversity at these locus. These traits, along with yield, were also rather negatively impacted with genetic diversity at the global genomic scale (Table 5).

To explain this reduction of the number of kernel per m$^2$ and yield when diversity increased at least on a part of the genome, one can remark that stem length showed a particular behaviour. While the stress reduced plant height, an interesting trend appeared at the genome global level: SNP whose diversity had a positive impact on stem length were in excess compared to those having a negative impact. This may reflect that plants increased their height as a reaction to the intra genomic complexity of their close environment. Plants increased their carbon allocation into stem and elongation in response to what they could have detected as a more competitive environment. This competition seemed to increase lightly under stress, diversity having a slightly stronger effect on plant elongation in CWD than in R. In our case, this happened despite the fact that we had restricted the variation in plant height and earliness among the 96 components. In addition, we found that diversity tended to elongate the spikes (SpL), as shown by a positive effect HE-QTL on chr 3B and a genomic excess of SNP whose diversity elongated spikes under R conditions. If spike length is usually expected to reflect the number of kernels per spike, it was not the case here (SpL and NKS were not correlated, Fig 2). Instead, we postulate that in our mixture condition, the spike elongation may be due to competition (and the positive correlation with stem length) but had no impact on spike fecundity. Instead, this competition-based increased carbon allocation into straw may have resulted in a reduction of the number of kernels per spike (NKS). Indeed, diversity was not favourable for this trait: the HE-QTL detected on the 1B chromosome had a negative impact on NKS in both water regimes and there were also more SNP whose HE had a negative impact than the opposite.

Unlike most components of yield, kernel weight was affected positively by diversity: we found a HE-QTL for TKW on chr 4A with a positive effect and the global effect was also positive (Table 5). These results are very probably linked with the existing trade-off existing between NKM2 and TKW (e.g. [84]). Under this assumption the positive response of TKW to genetic diversity is indirect and driven by the negative answer of NKM2 to the same cue.

**Ecological and evolutionary mechanisms involved.** Overall, our results disagree with the prediction that niche complementarity and facilitation should increase productivity when genetic diversity increases [24, 85]. A negative effect of genetic diversity at a single locus was already found in a previous study conducted on 197 two-way mixtures designed from the same durum wheat population [49]. Another study recently found a negative effect of genomic diversity at multiple loci on stand-level productivity in Arabidopsis thaliana communities [48] (Turner et al., 2020). Then, current results suggest that negative effects of diversity at the genomic level might not be so uncommon. Such negative effect of genomic diversity at single loci can be interpreted as a positive effect of genomic relatedness: a given allele has a higher reproductive output when it is grown with neighbours who share a similar allele. In evolutionary biology, interactions where changes in allele frequencies are driven by relatedness at a single locus have traditionally been investigated within the framework of "Greenbeard genes" [34]. Greenbeard genes could favour their own transmission by making individuals either more altruistic towards other individuals sharing the same gene copy or more harmful towards individuals bearing a different copy [34, 86]. If applied to crop mixtures, this effect would therefore predict that mixtures in which genotypes share the same allele at some locus are more productive than mixtures in which genotypes have different alleles [87]. Based on the green beard theory, two hypotheses could explain the negative effect of allelic diversity on yield: i) a negative plant x plant interaction between individuals carrying different alleles at the same locus, such as a competitive effect, ii) a positive interaction between individuals carrying identical alleles at the locus such a cooperative effect [88, 89]. So in durum wheat, diversity at some locus might oppose to a global positive effect of varietal diversity at other loci. Interestingly, the study by

Montazeaud [49] also reported a significant positive overyielding overall, suggesting that negative effect of allelic diversity found at a particular locus was not incompatible with an overall positive effect of varietal diversity. Unfortunately here, yield data were not available for the 96 pure components and we could not test if we had a global beneficial overyielding impact in mixtures, as found by [52].

The He-QTL on the chr 1B with a negative effect on the number of kernels per spike (NKS and Y) co-localizes with two published QTLs, one associated to grain yield [74] and another to test weight [71]. The QTL boundaries is of 36,229 bp in which we found a gene "TRITD1Bv1G013230" located in the [32,914,33 – 32,915,768] boundaries, and found responsible of the TOX high mobility group box protein, putative (DUF1635). In the literature, this gene has not a function identified in plants. This HE-QTL would be a particularly interesting candidate for a so-called green beard locus, whose diversity is therefore expected to reduce the performance of the group. It is possible that the HE-QTL on chr 1B includes a gene involved in one of the biochemical pathways implied in plant communication and recognition, such as the allelochemical DIMBOA, strigolactone or COV [90–92], but we have no candidate yet. It may be first interesting to confirm this HE-QTL by experimenting in mixtures with a simplified composition, to reduce the confidence interval of the QTL and to annotate more precisely the genome area.

Even if the interpretation of the mechanisms involved remains speculative, the effects of diversity detected here are consistent with previous observations showing that, beyond the average phenotypic value predicted from the values of pure components, plant x plant interactions are also at play and may explain part of the among mixture variation.

**Mixtures, diversity and stress.** One of the factors that could affect plant x plant interactions in mixtures, and explain the observed high variability in the effects of diversity, is environmental conditions. The stress gradient hypothesis [36] applied to agricultural conditions predicts that a stressful environment should increase the positive effects of diversity in mixtures compared to the standard conditions experienced by plants in intensive agriculture. Previous studies provided support for the stress gradient hypothesis in the context of crop mixtures, by showing that over-yielding increased under high disease pressure [29] or that mixtures were more tolerant to environmental stresses [93]. Among the mechanisms that might be responsible for these effects, stresses could foster positive interactions, such as complementarity and compensation between the varieties of the mixtures.

**Stress induced by the water deficit.** In our experiment, the controlled water deficit induced a measurable stress compared to the rainy conditions. Under water stress we observed effects that are consistent with previous results [94–96], such as smaller and earlier-flowering plants. Plasticity leading to early flowering time and a shorter vegetative phase is seen as an important and classical adaptive feature for wheat production under drought to minimize exposure to dehydration during the sensitive flowering and post-anthesis grain filling periods and to complete their life-cycle quickly during the brief period of favourable conditions [96, 97]. Water stress also affected the reproductive success of plants, diminishing slightly the number of spike per m$^2$. The water stress imposed at the end of tillering, when the number of spikes is determined [98, 99] was mild, but increased during the rest of the crop cycle, in particular during the stem elongation or jointing [95]. This may have resulted in a stronger regression of tillers, causing a final decrease of NSM2 under water stress. The stress then impacted the number of kernels per spike (NKS) during spike growth [99]. Yet, drought also depleted the kernel weight in spite of the lower competition for assimilates between kernels induced by the lower NKM2. Overall, this lead to a strong reduction in yield (Y), as commonly reported [100–102].

**Test of the stress gradient hypothesis.** For two HE-QTLs on yield (as well as NKM2), we found an interaction with the environment, and the slope was positive under stress conditions

only. These results are in agreement with the stress gradient hypothesis that predicts mixtures could benefit from their genetic diversity at some loci when growing in non-optimal abiotic conditions. We can say that positive interactions become more important and negative interactions become less important as the environment becomes harsher. Under stress, the diversity at these loci may reflect a phenotypic complementarity between the components within mixtures, leading to an improvement in the access to resources, water in particular [103–105]. A compensation mechanism may also explain the stress gradient hypothesis if some varieties are affected by the stress during a critical development phase while others escape it and eventually compensate partly the yield [106, 107].

At the global genetic level, however, we found that global genetic diversity was more detrimental under stressful conditions than under normal conditions for fitness related traits such as NKS, NKM2 and yield. If positive effects were observed on a small number of loci, the distribution was overall skewed towards negative impacts of diversity under stress (e.g. Fig 5). This is not in favour of the stress gradient hypothesis. Plant height was the only trait that validated the stress gradient hypothesis as the global genetic level, but as explained in the previous paragraph, this might be an effect of the perceived level of competition.

Overall, our experiment provides limited support for the stress gradient hypothesis, thereby suggesting that complementarity or compensation do not mitigate the effects of stress in our mixtures. Instead, competition is harsher in the stressful environment and negatively affects the yield. In her meta-analysis, Borg et al. (2018) were also unsuccessful in their attempt to test the influence of abiotic stress, due to lack of sufficient studies (not enough studies and not enough details about the nature of the stress) [29]. This underlines the need for further studies to better understand the ecological mechanisms that determine mixtures performances, in particular the relative roles of competition versus complementarity.

### Guidelines to assemble genotypes in a mixture

Our results highlight complex and contradictory mechanisms at play in our 96 mixtures. Finding variable effects in our experiment mirrors large variability in the over-yielding reported among mixtures in the literature. Finding significant associations between yield or its components and the diversity at specific loci is a methodological proof of concept that plant x plant interactions can shape the performance of crop mixtures. Even if there is an excess of negative associations between diversity and yield, a large number of loci still have a positive effect (Fig 5). Ultimately, assembling mixtures by selecting genotypes that differ on some loci for which there is evidence of complementarity or compensation, but remain uniform for others loci, such as candidate green-beard locus. This blind approach could complement trait based assembly rules [44].

### Supporting information

**S1 Fig. The distribution of plant height of the 96 EPOs selected from the 180 lines of EPOs extracted for our experimentation.** Red bars: 180 lines extracted from EPOs. Blue bars: 96 selected EPOs for the experiment, with uniform plant height.
(TIF)

**S2 Fig. The distribution of earliness of the 96 EPOs selected from the 180 lines of EPOs extracted for our experimentation.** Red bars: 180 lines extracted from EPOs. Blue bars: 96 selected EPOs for the experiment, with uniform plant height.
(TIF)

**S3 Fig. Genome wide association between allelic frequencies and TKW on a set of 96 12 way mixture and 96,582 bi allelic SNP.** (A) Manhattan plot of the GWFA of TKW and 96k SNPs distributed along the durum wheat genome. The solid blue line represents the Family-Wise Error Rate (FWER) of 5% computed with the Galwey method (with a value of 4.55). (B) Relation between the allelic frequency of the peak SNP "AX-89444977" and TKW. Red points corresponds to plots under controlled water deficit conditions. Black points corresponds to plots under rainy conditions.
(TIF)

**S1 Table. SNPs having significant associations between their allelic frequencies and the yield and others traits, with their–log10(PValue) and their positions on the chromosomes.** SpL: Spike Length in cm, TKW: Thousand Kernels weight in g, SNP: Single Nucleotide Polymorphism, Peak: physical position of the SNP on the chromosome in base pairs (bp), Lower bound: the minimal physical position that can have the SNP on the chromosome, Upper bound: the maximal physical position that can have the SNP on the chromosome, Chr: chromosome, QTL: Quantitative Trait Loci. The threshold of GWFA–log10(PValue) is 4.46. When GWFA–log10(PValue) is > 4.46 the association between the trait and the QTL is significant, and vice versa. * Significant–log10(PValue).
(DOCX)

**S2 Table. QTLs having significant associations between their genetic diversity, the yield, and its components with the sign of the effect of their genetic diversity on each trait and their positions on chromosomes.** SL: Stem length in cm, SpL: Spike Length in cm, NKS: Number of kernels per spike, NKM2: Number of kernels per $m^2$, TKW: Thousand Kernels weight in g, Y: yield in gr/$m^2$, SNP: Single Nucleotide Polymorphism, Peak: physical position of the SNP on the chromosome in base pairs (bp), Lower bound: the minimal physical position that can have the SNP on the chromosome, Upper bound: the maximal physical position that can have the SNP on the chromosome, Chr: chromosome, QTL: Quantitative Trait Loci. The threshold of GWFA–log10(PValue) is 4.46. * Significant test.
(DOCX)

**S3 Table. QTLs found that their allelic frequency is associated to traits, with their positions on the svevo genome of durum wheat, their–log10 (PValue) and the traits associated to their co-localized QTLs.** SpL: Spike Length in cm, TKW: Thousand Kernels weight in g, SNP: Single Nucleotide Polymorphism, Peak: physical position of the SNP on the chromosome in base pairs (bp), Lower bound: the minimal physical position that can have the SNP on the chromosome, Upper bound: the maximal physical position that can have the SNP on the chromosome, Chr: chromosome, QTL: Quantitative Trait Loci. The threshold of GWFA–log10 (PValue) is 4.46. * Significant test.
(DOCX)

**S4 Table. QTLs found that their genetic diversity is associated to traits, with their positions on the svevo genome of durum wheat, their–log 10 (PVal) and the traits associated to their co-localized QTLs.** SpL: Spike Length in cm, NKS: Number of kernels per spike, NKM2: Number of kernels per $m^2$, TKW: Thousand Kernels weight in g, SNP: Single Nucleotide Polymorphism, Peak: physical position of the SNP on the chromosome in base pairs (bp), Lower bound: the minimal physical position that can have the SNP on the chromosome, Upper bound: the maximal physical position that can have the SNP on the chromosome, Chr: chromosome, QTL: Quantitative Trait Loci. The threshold of GWDA–log10(PValue) is 4.46. The threshold of HE vs F test–log10(PValue) is 2.78. * Significant test.
(DOCX)

## Author Contributions

**Conceptualization:** Vincent Allard, Jacques L. David.

**Data curation:** Pauline Alsabbagh, Morgane Ardisson, Aline Rocher, Vincent Allard.

**Formal analysis:** Pauline Alsabbagh, Michel Colombo, Jacques L. David.

**Investigation:** Pauline Alsabbagh, Jacques L. David.

**Methodology:** Jacques L. David.

**Project administration:** Jacques L. David.

**Supervision:** Jacques L. David.

**Writing – original draft:** Pauline Alsabbagh.

**Writing – review & editing:** Laurène Gay, Germain Montazeaud, Vincent Allard, Jacques L. David.

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
