## [Decision Letter · Decision Letter 0]

31 Jan 2022

PONE-D-21-35051Diversity matters in wheat mixtures: a genomic survey of the impact of genetic diversity on the performance of 12 way durum wheat mixtures grown in two constrasted and controlled environments.PLOS ONE

Dear Dr. Alsabbagh,

Thank you for submitting your manuscript to PLOS ONE. After careful consideration, we feel that it has merit but does not fully meet PLOS ONE’s publication criteria as it currently stands. Therefore, we invite you to submit a revised version of the manuscript that addresses the points raised during the review. Please submit your revised manuscript by Mar 17 2022 11:59PM. If you will need more time than this to complete your revisions, please reply to this message or contact the journal office at plosone@plos.org. Please include the following items when submitting your revised manuscript:A rebuttal letter that responds to each point raised by the academic editor and reviewer(s). You should upload this letter as a separate file labeled 'Response to Reviewers'.A marked-up copy of your manuscript that highlights changes made to the original version. You should upload this as a separate file labeled 'Revised Manuscript with Track Changes'.An unmarked version of your revised paper without tracked changes. You should upload this as a separate file labeled 'Manuscript'.

We look forward to receiving your revised manuscript.

Kind regards,

Ajay Kumar

Academic Editor

PLOS ONE

Journal Requirements:

“The EU H2020 SOLACE funded the experiment; PA received a grant from l’Assocation de Spécialisation et d’Orientation Scientifique du Liban.”

We note that you have provided additional information within the Funding Section that is not currently declared in your Funding Statement. Please note that funding information should not appear in other areas of your manuscript. We will only publish funding information present in the Funding Statement section of the online submission form.

“JD was funded by the EU Horizon2020 SOLACE grant number 727247.

PA recieved a grant from "l'Association de spécialisation et d'orientation scientifique, Liban".”

“JD was funded by the EU Horizon2020 SOLACE grant number 727247.

PA recieved a grant from "l'Association de spécialisation et d'orientation scientifique, Liban".”

Reviewers' comments:

Reviewer's Responses to Questions

**Comments to the Author**

1. Is the manuscript technically sound, and do the data support the conclusions?

Reviewer #1: Partly

Reviewer #2: Yes

2. Has the statistical analysis been performed appropriately and rigorously? 

Reviewer #1: No

Reviewer #2: Yes

3. Have the authors made all data underlying the findings in their manuscript fully available?

Reviewer #1: Yes

Reviewer #2: No

4. Is the manuscript presented in an intelligible fashion and written in standard English?

Reviewer #1: No

Reviewer #2: Yes

5. Review Comments to the Author

Reviewer #1: In general, I think this study is good and should be published. However, the writeup needs to be revised, requiring enhancement of the 2nd figure and editing to the writing (especially the introduction). The sentences used are too long and complex. Please pay attention to the comments i made in the PDF file.

Reviewer #2: The manuscript by Alsabbagh et al. is a description of a project to investigate the impact genetic diversity has on agronomic and yield-component traits. The authors generated 96 different 12-way mixtures and evaluated plot performance under normal and water-deficient treatments. They used genome-wide genotyping and regression techniques to identify if allele frequency and diversity of SNPs (and LD bocks) where associated with traits. The methodology is rigorous, and an explicit effort was made to decrease the Type 1 errors. Particularly, using a LD-derived method to determine QTL boundaries, instead of an arbitrary value is appreciated. However, the design is generally underpowered with only a single year of data and 48 unbalanced experimental units per treatment. This is partly compensated by the use of a well-characterized source population and careful mixture design, but at the end of the day, only year-specific large effects are discoverable with this design. This is plainly seen in figures 3-5 where single data points at extreme values are skewing regression lines. Are these SNPs still identified as significant if that single point is removed? (See below) That being said, publication would still benefit the community as this type of research is generally not see in the literature and this can serve as an anchor to future projects.

Generally, the manuscript is well-written, but a few areas need to be addressed. There are a few typos or grammatical errors identified on lines 192, 243, 338, 546, 620. Additionally, there are some sentences that were awkward and should be revised to improve clarity on lines 347, 661. Informal language is used on lines 311, 577, and 598 that should be revised. On line 584, “proves” is not an appropriate term.

In the materials and methods, it is not clear how SNPs were filtered for the GWDA. From the first sentence, I gathered that SNPs needed to exhibit a min and max frequency value of 0.25-0.75. However, I don’t think that’s correct and the given example does not assist the reader to understand this. On line 294, it states that 13,093 markers were left, but then states 31,059 in the “Genotypic Data” Section, which one is it?

All the figures and subpanels need legends and axis titles included so the reader doesn’t need the read the caption to understand what is going on. As stated above, it looks like the regression lines are influenced from extreme values, this needs to be checked. Perhaps an additional filtering step to remove top and bottom tails of the frequency and diversity distribution should be added to eliminate the presumed spurious associations. Additionally, Figures (3&4)c show phenotype values from both treatments co-plotted by diversity and a single regression line. This is misleading because there should have been a treatment effect included with the determination of this regression slope. So either, you need to show both lines (as in Fig5) or plot the BLUEs versus the single line.

6. PLOS authors have the option to publish the peer review history of their article (what does this mean?). If published, this will include your full peer review and any attached files.

Reviewer #1: No

Reviewer #2: No

---

## [Author Response · Author response to Decision Letter 0]

16 Aug 2022

Thank you for your time and effort to revise my manuscript. I revised the manuscript of the article and made improved the text based on your comments.

---

## [Decision Letter · Decision Letter 1]

4 Oct 2022

Diversity matters in wheat mixtures: a genomic survey of the impact of genetic diversity on the performance of 12 way durum wheat mixtures grown in two constrasted and controlled environments.

PONE-D-21-35051R1

Dear Dr. Alsabbagh,

We’re pleased to inform you that your manuscript has been judged scientifically suitable for publication and will be formally accepted for publication once it meets all outstanding technical requirements.

Kind regards,

Mehdi Rahimi, Ph.D.

Academic Editor

PLOS ONE

Additional Editor Comments (optional):

Reviewers' comments:

Reviewer's Responses to Questions

**Comments to the Author**

1. If the authors have adequately addressed your comments raised in a previous round of review and you feel that this manuscript is now acceptable for publication, you may indicate that here to bypass the “Comments to the Author” section, enter your conflict of interest statement in the “Confidential to Editor” section, and submit your "Accept" recommendation.

Reviewer #1: All comments have been addressed

Reviewer #2: All comments have been addressed

2. Is the manuscript technically sound, and do the data support the conclusions?

Reviewer #1: Yes

Reviewer #2: Yes

3. Has the statistical analysis been performed appropriately and rigorously? 

Reviewer #1: Yes

Reviewer #2: Yes

4. Have the authors made all data underlying the findings in their manuscript fully available?

Reviewer #1: Yes

Reviewer #2: Yes

5. Is the manuscript presented in an intelligible fashion and written in standard English?

Reviewer #1: Yes

Reviewer #2: Yes

6. Review Comments to the Author

Reviewer #1: The author have addressed the comments made last time. However i did notice some formatting issues. For eg. Line 467 - 483 is single spaced and Figure 1 has different font colors ( probably because of the program used to generate the figures). I request the author to fix for consistency.

Reviewer #2: Revisions have been addressed with proper annotation of changes in the revised manuscript. The figure resolution on my copy is quite poor. Please ensure publication-ready figures are supplied.

7. PLOS authors have the option to publish the peer review history of their article (what does this mean?). If published, this will include your full peer review and any attached files.

Reviewer #1: No

Reviewer #2: No

---

## [Editor Report · Acceptance letter]

11 Nov 2022

PONE-D-21-35051R1 

Diversity matters in wheat mixtures: a genomic survey of the impact of genetic diversity on the performance of 12 way durum wheat mixtures grown in two constrasted and controlled environments. 

Dear Dr. David:

I'm pleased to inform you that your manuscript has been deemed suitable for publication in PLOS ONE. Congratulations! Your manuscript is now with our production department. 

Kind regards, 

on behalf of

Dr. Mehdi Rahimi 

Academic Editor

PLOS ONE